# Female Cuckoo Calls Deceive Their Hosts by Evoking Nest-Leaving Behavior: Variation under Different Levels of Parasitism

**DOI:** 10.3390/ani12151990

**Published:** 2022-08-05

**Authors:** Jiaojiao Wang, Laikun Ma, Xiangyang Chen, Canchao Yang

**Affiliations:** 1Ministry of Education Key Laboratory for Ecology of Tropical Islands, College of Life Sciences, Hainan Normal University, Haikou 571158, China; 2College of Life Science, Hebei University, Baoding 071000, China; 3Department of Biology and Food Science, Hebei Normal University for Nationalities, Chengde 067000, China

**Keywords:** anti-parasite strategy, anti-predator behavior, avian brood parasitism, parasitic strategy, vocal mimicry

## Abstract

**Simple Summary:**

Obligate brood parasite birds such as the common cuckoo (*Cuculus canorus*) can trick their hosts via a variety of means. Cuckoos are threats to the host nests but not to the host adults themselves. To successfully parasitize the host nests, female cuckoos have been hypothesized to distract the hosts’ attention from their nests by mimicking the calls of sparrowhawks (*Accipiter nisus*), a predator of the hosts. We performed playback experiments in two populations of the host oriental reed warbler (*Acrocephalus orientalis*) that have experienced different levels of parasitic pressure. We found that female cuckoo calls evoked both populations of the hosts to leave their nests more frequently than did the calls of male cuckoos or doves that do not pose threats to the hosts. This indicated that the call of the female cuckoo functions to deceive the host and thus favors host brood parasitism. However, we propose that such a deceiving effect of the female cuckoo call is due to the rapid cadence of the call rather than sparrowhawk mimicry.

**Abstract:**

The common cuckoo (*Cuculus canorus*) is an obligate brood parasite that has evolved a series of strategies to trick its hosts. The female cuckoo has been hypothesized to mimic the appearance and sounds of several raptors to deceive the hosts into exhibiting anti-predator behavior. Such behavior would relax the protection of the host nest and thus allow the female cuckoo to approach the host nest unopposed. Many anti-parasite strategies have been found to vary among geographical populations due to different parasitic pressures from cuckoos. However, the effect of female cuckoo calls related to different levels of parasitic pressure has not been examined. Here, we studied the effect of female cuckoo calls on the oriental reed warbler (*Acrocephalus orientalis*), one of the major hosts of the common cuckoo, in two geographical populations experiencing different levels of parasite pressure. Four kinds of sounds were played back to the hosts: the calls from female common cuckoos, male common cuckoos, sparrowhawks (*Accipiter nisus*), and oriental turtle doves (*Streptopelia orientalis*). The results showed that the female cuckoo calls induced the hosts to leave their nests more frequently than the male cuckoo or dove calls in both populations, and two populations of the hosts reacted similarly to the female cuckoo calls, implying that the function of female cuckoo calls would not be affected by the difference in parasitism rate. This study indicates that female cuckoo calls function to distract the hosts’ attention from protecting their nests. However, we propose that such a deception by the female cuckoo call may not be due to the mimicry of sparrowhawk calls, but rather that the rapid cadence of the call that causes a sense of anxiety in the hosts.

## 1. Introduction

In the coevolution of host and brood parasitic birds such as the common cuckoo (*Cuculus canorus*), many specialized parasitic and anti-parasite strategies have evolved [1,2]. The cost of parasitism can be radically reduced by the host preventing parasitic birds from entering the nest to lay eggs [3,4]. Therefore, the hosts have evolved the ability to recognize the parasites and perform various nest defense and attack behaviors, causing different degrees of harm to the parasites [5,6,7,8]. Some aggressive hosts such as the great reed warbler (*Acrocephalus arundinaceus*) can even kill cuckoos [9,10]. In addition, after observing a cuckoo, the host may enhance the intensity of inspecting the nest, thereby increasing the recognition and rejection rates of parasitic eggs [11]. These nest defense behaviors of the hosts have prompted further optimization of parasitic strategies during coevolution.

In addition to remaining concealed during laying [12], some parasitic birds employ strategies that visually mimic raptors [13,14]. The most well-known example is the Batesian mimicry of the common cuckoo of the shape of a raptor that, at least from a human perspective, is highly similar to that of the sparrowhawk (*Accipiter nisus*) [15]. Cuckoos pose a different threat to adult host birds than sparrowhawks. The former are physically harmless to adult host birds [16,17] while the latter are predators of many adult birds [18]. Thus, the cuckoo’s simulation of a sparrowhawk can deceive the host and trigger the host’s anti-predator response. This is beneficial for the cuckoo, as it can increase the rate of parasitism [13]. Currently, studies have verified that cuckoos can visually mimic raptors [7,13,14,19,20,21,22,23,24], but there is much less research on vocal mimicry. Recent studies have shown that female calls, which are bubbling calls, can also mimic sparrowhawk sounds, thus playing a similar deceptive role in physical mimicry [25,26]. This response to auditory mimicry has been verified in birds including wild free-range chickens (*Gallus domesticus*), cinereous tits (*Parus cinereus*), and the oriental magpie-robin (*Copsychus saularis*) [27,28,29]. In addition, Marton et al. [30] found that the calls of female cuckoos reduced the attack intensity of great reed warblers in experiments combining sound playback and 3D-printed models, while similar experiments on non-host yellow-rumped flycatchers (*Ficedula zanthopygia*) yielded contrasting results [28]. However, Deng et al. [31] and Yoo et al. [32] found that the song peak of female cuckoos was inconsistent with the time of laying. Gong et al. [33] showed that the song peaks of female cuckoos occurred at sunrise and sunset. These studies suggest that the calls of female cuckoos may be used for purposes other than to deceive hosts. The bubbling female cuckoo call is a multifunction call. Some studies have suggested other functions for the bubbling call of female cuckoos, such as for advertising females’ laying territories, thus reducing female–female aggression [34,35,36], as well as for mate attraction [34,35], or male–female duetting [37]. When female cuckoo calls are used to dampen host aggression at host nests, the number of calls they make may be much lower than when they perform other functions.

In brood parasitism, different host populations are subjected to varied parasitic pressures due to the differences in habitat, population density, the presence or absence of parasites, and the number of parasites [38,39,40]. This may lead to different anti-parasite strategies of the hosts. For example, the mobbing rate of reed warblers (*A. scirpaceus*) against the common cuckoo decreased as the parasitism rate decreased [41]; the superb fairywren (*Malurus cyaneus*) can distinguish cuckoos with different degrees of parasitic risk, and there are more alarm calls in response to cuckoos with higher parasitic risk [42,43]. Similarly, in terms of egg stage defense, the host egg recognition ability of different populations also varies. For example, the European barn swallow (*Hirundo rustica*) cannot recognize foreign eggs, but the Chinese population has a strong ability to reject alien eggs, and there are clear dimensional differences and geographic variation [44,45]. Therefore, the effectiveness of cuckoo simulation of sparrowhawks also needs to be verified in different host populations.

In this study, we selected two populations of the oriental reed warbler (*Acrocephalus orientalis*) experiencing different levels of parasitic pressure and played back four types of calls (the female common cuckoo, the male common cuckoo, the sparrowhawk, and a harmless control) to investigate the function of female cuckoo calls and the variation between two host populations with different levels of parasitism. We hypothesized that: (1) If the calls of female cuckoo and sparrowhawk have similar functions, the two populations of oriental reed warbler should exhibit similar behaviors toward the calls of the female cuckoo and sparrowhawk, and the behaviors would significantly higher than they toward the calls of male cuckoo and dove, which may be independent of the local parasitism rate. (2) If the function of female cuckoo calls is not to reduce host aggression, but to protect the laying territories or attract mates, then the response of the two populations to female cuckoo calls should be similar to their response of male cuckoo or dove, which may be independent of the parasitism rate of the two populations. (3) If the calls of female cuckoo do not mimic the sparrowhawk, but since the cuckoo and sparrowhawk are different types of predators, the female calls may cause a similar response as the sparrowhawk’s calls, and then this may be influenced by the parasitism rate and predation rate of the two regions, or the female cuckoo calls may have other functions.

## 2. Materials and Methods

### 2.1. Study Site and Subjects

The research areas were located in Yongnianwa National Wetland Park of Hebei Province (36°40′–36°41′ N, 114°41′–114°45′ E) and Zhalong National Nature Reserve of Heilongjiang Province (46°48′–47°31′ N, 123°51′–124°37′ E). Reeds (*Phragmites australis*) and cattails (*Typha latifolia*) are the main plants in both wetlands [46]. The cuckoo is the most common obligate interspecific brood parasitic bird in Europe and Asia [47,48]. The oriental reed warbler is one of the main hosts of the cuckoo, and the coevolution between the two species has been proposed to have reached a high level [49,50,51,52]. Oriental reed warblers are distributed in both Yongnian and Zhalong areas, but the probability of being parasitized by cuckoos differs. In Yongnian, a total of 257 oriental reed warbler nests were found during the breeding seasons 2016–2017, with an overall parasitism rate of 14.8% [46]. However in Zhalong, the probability of oriental reed warbler being parasitized by the common cuckoo was 13% in 2012 (*n* = 73 host nests) [49] and 65.5% in 2013 (*n* = 55) [53]. Previous studies on oriental reed warblers in Yongnian through specimen experiments showed that the host birds could visually distinguish cuckoos from sparrowhawks and doves [54]. Replay experiments employing the conspecific alarm calls of oriental reed warblers in Zhalong have shown that the warblers responded more strongly to alarm calls toward cuckoos than to alarm calls toward sparrowhawks [55], suggesting that the warbler can visually distinguish cuckoos from predators.

### 2.2. Sound Production and Playback

The songs of male and female cuckoos, sparrowhawks, and oriental turtle doves (*Streptopelia orientalis*) were used as different types of playback stimuli. The cuckoos are nest parasites that threaten the host nests but not the host adults; the sparrowhawks are predators that threaten the host adults, while the doves are harmless controls. The sounds were downloaded from XenoCanto recordings (http://www.xeno-canto.org/, accessed on 3 May 2017) (Figure 1). Similar to the study of Shen et al. [28], we also used sparrowhawk’s other call types with a lower call rate than the study of York and Davies [25]. Two samples of each type of sound were combined for the playback experiments, and two minutes were used for each playback sound. All four types of sounds were played back at each nest. From May to July in 2017, 2020, and 2021, the four sounds were played back randomly to oriental reed warblers during egg incubation (Yongnian: *n* = 22, days after clutch completion: 4.05 ± 2.46 (mean ± SD) days; Zhalong: *n* = 23, days after clutch completion: 3.17 ± 3.05 (mean ± SD) days). The specific operation steps were as follows. We installed micro cameras approximately 30 cm above the nest and placed the player (BV370, SEE ME HERE Electronic Corporation, Shenzhen, China) approximately 1 m away from the nest. An observer hid in the reeds approximately 10 m away, waited for two minutes after the host parent bird had returned to the nest and begun incubating, and then began to play the sounds. The playback lasted for two minutes. Each nest had the above four sounds played back in random sequence, and the playback interval between the two sounds was at least 15 min. The following behaviors of brooding parents were recorded: (1) stay on or leave the nest, and (2) latency to leaving (the latency time was recorded only for the nests whose brooding parents really leave).

### 2.3. Statistical Analyses

Cox regression controlling for nest identity was used to calculate the probability of staying on nests by hosts after playback, and the probability was compared between different playback stimuli and host populations [56]. In the regression model, the event consisted of staying on or leaving nests, where the latter refers to the occurrence of the event, while the latency to leaving is the time. The tested effects included the stimuli (four types of playback), population (Yongnian or Zhalong), clutch size, incubation day (the day of incubation when the experiment was performed), egg laying date (the date of the first laid egg), and the interaction between population and stimuli. Cox regression was also used for pairwise comparisons between playback stimuli for each host population. A Kaplan–Meier curve with a 95% confidence interval was generated to visualize the probability of staying on nests by hosts during the playback experiment. The Cox regression and Kaplan–Meier curve were constructed by using the *survival* and *survminer* packages in R (Version 4.1.0) for Windows (https://www.r-project.org/, accessed on 20 April 2021). All statistical tests were two-tailed, with a significance level of *p* < 0.05.

## 3. Results

The responses of oriental reed warbler to the different playback callsin the two regions are shown in Table 1.

There was no significant difference in the response toward playback stimuli between Yongnian and Zhalong populations (Z = 0.191, *p* = 0.191, Cox regression), but there were significant differences for the playback stimuli (Z = −2.672, *p* < 0.01, Cox regression) and the interaction between stimuli and population (Z = −2.267, *p* < 0.01, Cox regression; Table 2). This indicated that the hosts reacted differently to different playback stimuli, and the reaction also differed between populations. Pairwise comparisons indicated that there was no significant difference between hosts’ responses to sparrowhawk and female common cuckoo calls in either Yongnian (Z = 0.020, *p* = 0.984, Cox regression) or Zhalong (Z = −1.514, *p* = 0.130, Cox regression) populations. Similarly, there was no significant difference between the responses to dove and male common cuckoo calls for either population (Yongnian: Z = −0.532, *p* = 0.594; Zhalong: Z = −0.049, *p* = 0.961, Cox regression). Furthermore, the comparison results between female and male cuckoo calls and between female cuckoo and dove calls were also consistent between Yongnian and Zhalong populations (Table 3), where significant differences were detected for both (*p* = 0.030 and 0.014 for Yongnian and *p* = 0.023 and 0.033 for Zhalong, respectively; Cox regression).

However, the results of comparisons between male cuckoo and sparrowhawk calls showed opposite patterns between the two populations (Yongnian: *p* = 0.037; Zhalong: *p* = 0.309, Cox regression). Similarly, the hosts’ responses to sparrowhawk and dove calls were also inconsistent between Yongnian and Zhalong populations (*p* = 0.020 and 0.312, respectively, Cox regression; Table 3). The Kaplan–Meier curve revealed more details of such inconsistent results. The female cuckoo calls in both populations triggered stronger reactions from hosts than the male cuckoo or dove calls because the female cuckoo calls significantly reduced the hosts’ probability of staying on the nest (Figure 2). However, for the sparrowhawk call, its effect in Zhalong was similar to those of male cuckoo and dove calls, contrary to the effect in Yongnian (Figure 2).

## 4. Discussion

The results showed that the oriental reed warblers in the Yongnian (lower parasitism rate) and Zhalong (higher parasitism rate) populations had similar overall responses to playback stimuli, but there were differences in specific responses to different types of calls. The female cuckoo calls in both populations caused the hosts to leave their nests significantly more frequently than the male cuckoo and dove calls, suggesting that the female cuckoo call serves to mislead the host during incubation. The sparrowhawk call elicited similar responses as the female cuckoo call in the Yongnian population. However, in the Zhalong population the effect was similar to male cuckoo and dove calls, without significant differences. These results suggest that the hosts are sensitive to female cuckoo calls in both Yongnian and Zhalong populations, but the Zhalong population is less sensitive to the sparrowhawk call than the Yongnian population.

The diversified anti-parasite strategies of hosts may have prompted cuckoos to evolve more efficient parasitic strategies. The “sparrowhawk-like” mimicry of body appearance for common cuckoos has been examined in several studies, e.g., [19,20,21,23,57,58]. Female common cuckoos can significantly reduce host fitness through brood parasitism without threatening the adult birds [1,2]. In contrast, the sparrowhawk is an important predator that can directly threaten adult passerine birds [18]. If the female cuckoo call can evoke the host anti-predator behavior similar to that displayed when hearing the call of the sparrowhawk, this would benefit the parasitic birds [26]. In recent years, studies have verified that female cuckoo calls simulate sparrowhawk calls and can deceive hosts, but these studies have not considered different geographic populations [25,28,29]. Different geographical populations may exhibit different anti-parasite strategies due to variation in parasite pressure [42,43]. In this study, four types of calls (female cuckoo, male cuckoo, sparrowhawk, and dove) were played back to oriental reed warblers in two populations experiencing different levels of brood parasitic pressure. The parasitism rate in Zhalong (up to 65.5%) was much higher than in Yongnian (14.8%) [46,59]. Similar to York and Davies [25] and Shen et al. [28], we conducted our experiment during the host’s incubation period. Although female cuckoo bubbling calls may be used during the laying period of the host, the host stays in the nest for too short a time for playback experiments to be meaningful. However, different from our first hypothesis, in both populations the female cuckoo calls caused the hosts to leave their nests more frequently than the male cuckoo or dove calls. This indicated that the hosts in both populations were misled by the female cuckoo calls. The two populations of the hosts reacted similarly to the female cuckoo calls, implying that the function of female cuckoo calls would not be affected by the difference in parasitism rate. Nevertheless, the sparrowhawk call triggered a higher probability of leaving the nest than the harmless control in Yongnian but not in Zhalong. This suggested that the predation risk from sparrowhawks may be higher in Yongnian than in Zhalong. Further studies are needed to verify this hypothesis by investigating the predation risk from sparrowhawks in these two populations.

Although the main conclusion (i.e., the misleading function of female cuckoo calls) in this study is similar to those from several previous studies [25,27,28], it cannot be concluded that the hosts left their nests after hearing the female cuckoo call because they misidentified it as a sparrowhawk call. Actually, according to our acoustic data, it is unlikely that the female cuckoo call evokes hosts’ leaving behavior because it mimics a sparrowhawk call. First, the reaction to female cuckoo calls was more consistent than to sparrowhawk calls between different populations. Second and importantly, the sonogram of female cuckoo calls is very different from that of the sparrowhawk call in the shape, frequency, and rate (Figure 1). The female cuckoo call is much more rapid. Therefore, we propose that the female cuckoo call can evoke the nest-leaving behavior because its rapid cadence may cause anxiety in the hosts. When we played back the reduced call rate of female common cuckoo to the oriental reed warblers, they did not respond (unpublished data). Further research is needed to test this hypothesis. In addition, the differences in specific song parameters between female cuckoo and sparrowhawk calls or other birds with similar song structures need to be further studied.

The female cuckoo bubbling call is considered to have the function of defending laying territories or attracting mates [34,35,36], but our results found that the response of the oriental reed warbler after hearing it was inconsistent with the other two harmless sounds (contrary to our second hypothesis), which may be because the other function of female cuckoo calls is to trick and mislead the host, thus changing the host’s behavior. Female cuckoo calls can be used as a signal of external parasitic risk [3], a warbler may think that hearing a call near their nest means that it is at risk of being parasitized, so they leave their nests to distract the parasites. Another reason for this is that female cuckoo is also a predator of the host nest, and will destroy host nests that are not suitable for parasitism [60] to improve their chances for future parasitism. It is also reasonable for oriental reed warblers to leave the nest to protect their offspring. This may be to attack the cuckoo, but the specific motivation of the host after leaving the nest is not clear, and further research is needed. In summary, our results may support our third hypothesis. York and Davies [25] first proposed the hypothesis that the female common cuckoos simulated sparrowhawks using sound in the brood parasitism system, and then further verified this by comparing the responses of host and non-host species [27,28,29]. However, Jiang et al. [27] and Zhang et al. [29] conducted experiments in the non-breeding periods of birds, and Shen et al. [28] conducted a study on hole-nesting bird species during the breeding period. Additionally, only vigilance or flight responses were recorded during playback of female cuckoo calls of these studies [25,27,28,29]. Consistent with our research, York and Davies [25] also studied common cuckoo hosts at the breeding stage in reed habitats, but in this study, the oriental reed warbler responded more intensely to the female cuckoo calls by fleeing from their nests. The difference was probably due to the longer playback time used in our study. In our study, the playback time was two minutes (120 s), while York and Davies [25] only played the sound for approximately 3 s, suggesting that the shorter duration of playback may not be sufficient to trigger the leaving behavior of the hosts. Finally, the response to male cuckoo calls did not differ from the response to dove calls in either population. This is reasonable, as the male cuckoo does not engage in parasitism, and the male cuckoo call is primarily related to territory defense [35,61].

## 5. Conclusions

In summary, our research on the oriental reed warbler showed that female cuckoo calls can evoke the hosts’ nest-leaving behavior, as the calls are similar to sparrowhawk calls. However, the effect of female cuckoo calls on hosts may be related to the rapid cadence rather than simulation of a sparrowhawk call. It is necessary to verify the mechanism of deception in female cuckoo calls in future studies.

## Figures and Tables

**Figure 1 animals-12-01990-f001:**
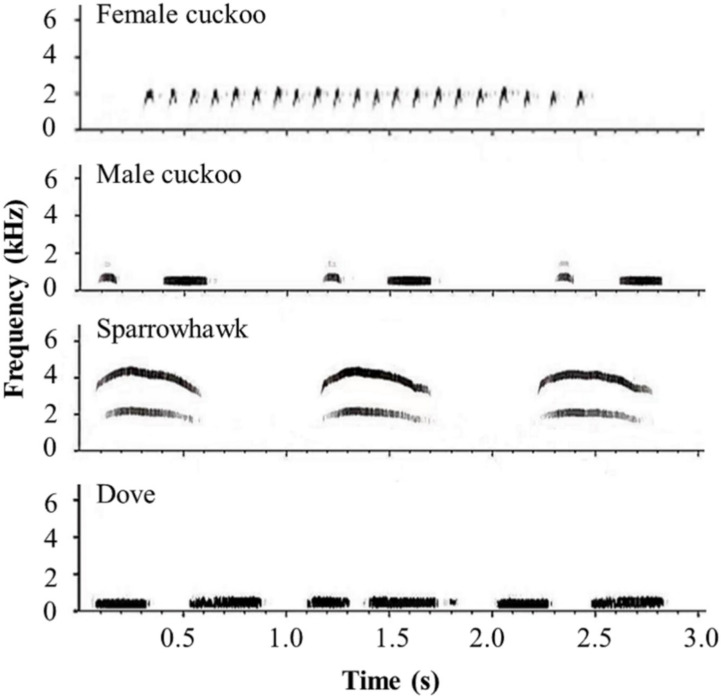
The spectrograms of four call types used in the playback experiments.

**Figure 2 animals-12-01990-f002:**
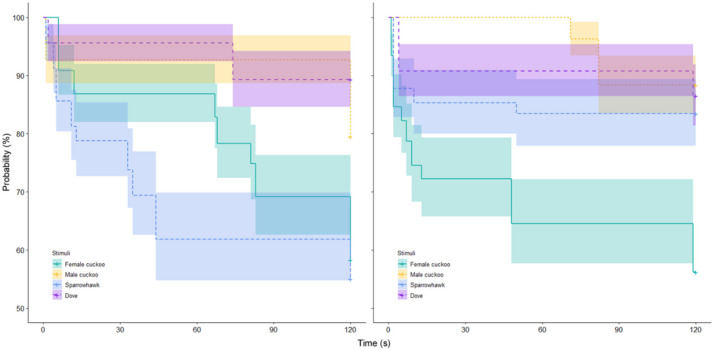
Kaplan–Meier curve (with 95% CI) for the probability of staying on nests by host parents from the playback of stimuli in Yongnian (**left**) and Zhalong (**right**) populations (the vertical axis represents the probability of the host staying in the nest, and the lower the value, the more likely the host is to leave the nest after hearing that specific sound).

**Table 1 animals-12-01990-t001:** Response of oriental reed warbler to different playback stimuli in two areas.

Areas	Playback Stimulus	Number of Nests the Host Did Not Leave	Number of Nests Left by Host	Latency to Leaving (s)
Yongnian	Female cuckoo	11	10	68.30 ± 46.34
	Male cuckoo	16	3	80.33 ± 68.70
	sparrowhawk	11	9	27.75 ± 39.43
	Dove	18	2	38.00 ± 50.91
Zhalong	Female cuckoo	14	9	22.78 ± 38.93
	Male cuckoo	20	2	76.50 ± 7.78
	sparrowhawk	18	4	16.00 ± 22.98
	Dove	20	2	61.50 ± 81.32

**Table 2 animals-12-01990-t002:** The result of Cox regression controlling for nest identity in this study.

Effects	Coefficient	SE	Z	*p*
Stimuli	−0.399	0.149	−2.672	<0.01 **
Population	−0.445	0.340	−1.307	0.191 ^ns^
Stimuli × Population	−0.267	0.090	−2.965	<0.01 **
Clutch size	0.053	0.206	0.258	0.797 ^ns^
Egg laying date	0.005	0.011	0.495	0.621 ^ns^
Incubation day	0.056	0.065	0.8856	0.392 ^ns^

^ns^*p* ≥ 0.05; ** *p* < 0.01.

**Table 3 animals-12-01990-t003:** The result of pairwise comparison for playback stimuli by Cox regression.

	Female Cuckoo	Male Cuckoo	Sparrowhawk	Dove
Yongnian population				
Female cuckoo		**0.030 ***	0.984 ^ns^	**0.014 ***
Male cuckoo	(−2.167)		**0.037 ***	0.594 ^ns^
Sparrowhawk	(0.020)	(2.084)		**0.020 ***
Dove	(−2.452)	(−0.532)	(−2.323)	
Zhalong population				
Female cuckoo		**0.023 ***	0.130 ^ns^	**0.033 ***
Male cuckoo	(−2.267)		0.309 ^ns^	0.961 ^ns^
Sparrowhawk	(−1.514)	(1.016)		0.312 ^ns^
Dove	(−2.132)	(−0.049)	(−1.012)	

Values in brackets are the statistics (Z values), while others are *p* values (^ns^
*p* ≥ 0.05; * *p* < 0.05). The *p* values with inconsistent significance between populations are shown in bold.

## Data Availability

The data presented in this study are available on request from the corresponding author.

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
