# Peer review of "Female Cuckoo Calls Deceive Their Hosts by Evoking Nest-Leaving Behavior: Variation under Different Levels of Parasitism"

_animals, 2022, doi:10.3390/ani12151990_

Round 1

Reviewer 1 Report

This study investigated common cuckoo females’ vocalization from the points of view of their hosts, the Oriental reed warblers. Cuckoo females have a special, multifunction call, the bubbling call, which is similar to sparrowhawks’ calls, and cuckoos may use this call to misinform hosts on the type of threat. The present study focused on hosts’ reactions to bubbling calls in two populations, with low and high levels of cuckoo parasitism. Despite the structural similarity of these two calls, Oriental reed warblers’ reactions proved to be different toward the two calls. It is a novel and interesting result. I agree with most parts of the experimental design, analyses and the manuscript, although I have several minor comments on the text.

You should clearly state that female cuckoo’ call, the bubbling call, is a multifunction call, it is not restricted to interspecific context in cuckoo-host relationships. This probably affects and compromises adaptations in different relations, and the bubbling call is the output of these adaptations. Studies in the last few years revealed that this multifunction call is used for:

(i) dampening host aggression at host nests, e.g., at laying attempts (York and Davies, 2017 Nat Ecol Evol; Marton et al., 2021 Ethology),

However, daily pattern of bubbling calls uttered suggested other main function of this call than (i) (Deng et al. 2019).

(ii) advertising females’ laying territories and, consequently, reducing female-female aggression (Moskát and Hauber 2019 BEAS, Xia et al. 2019 Ethology, Moskát and Hauber 2020 Sci Nat), and

(iii) male-female communication, like mate attraction (Moskát and Hauber 2019 BEAS, Xia et al. 2019 Ethology), and male-female duetting (Moskát and Hauber 2021 Behav Process).

The present study is related to function (i). For this reason, cuckoo females do not utter it frequently, as it is used when a female cuckoo lay one egg into a host nest. As cuckoos lay on every second day, it means that a cuckoo female uses the bubbling call for this purpose once in a two-day period (i.e., 0.5 call/day, or slightly more, if a female repeats this call a few times just after laying an egg into the host nest). For the other two purposes cuckoo females use the bubbling call more frequently. For example, sometimes in the peak period (ca. at the beginning or middle part of the laying period), certain females may call every 2-3 min, at a maximum. These suggest that you should also mention the other functions of it in the Introduction and Discussion because of their relevancy.

I understand that you wanted to compare the bubbling call and the more or less similar sparrowhawk call on host behaviour, also using common cuckoo males’ calls and dove calls (controls). Although several other papers also used this protocol, it is a somewhat unnatural situation, as cuckoo females, probably, never use their bubbling calls toward hosts when their nests are in the incubation stage. They lay their eggs in host nests during the laying period (i.e., in the pre-incubation period). However, in the laying period host females stay only short time on their nests, consequently, such experiments could not be conducted in the pre-incubation period. When you discuss your results, you have to take care of these details.

On the mimicry of bubbling calls to sparrowhawk calls. The two calls seem to be similar to each other, although no acoustic (statistical) analysis has already been carried out on the structural comparison of these two calls. The problem is that several bird calls or song fragments are also similar to these calls (little grebe, green woodpecker, etc.). I would eager to see a study which compares the signalling values of several similar call types with the sparrowhawk and bubbling calls.

            Another problem is that common cuckoos belong to the group of bird species called vocal non-learners. This means that they cannot learn any new sound types, which makes difficult their adaptation to a foreign call. Alternatively, I cannot exclude the possibility as female cuckoo, sparrowhawk, green woodpecker and little grebe calls could be similar as the consequence of other, presently unrevealed, acoustical constraints.

Minor comments

Line 56: The term "spawning" is typically applied for fishes, not for birds.

Line 65: Yes, bubbling call probably mimics sparrowhawk calls, as common cuckoos' plumage mimics sparrowhawk appearance (although cuckoo females have two colour variants, and the hepatic form does not mimic sparrowhawk).

Line 73: Use the term laying, instead of spawning!

The last paragraph of Introduction: Please, clarify hypotheses and predictions better.

Line 109: "The oriental reed warbler belongs to the family Acrocephalidae, Passerifirmes; the common cuckoo belongs to the family Cuculidae, Cuculiformes." I do not think this sentence is necessary here, it is more suitable for a thesis.

Line 110: using brood parasitism instead of nest parasitism would be better (throughout the text)

122: It would be necessary to know more details on the nests used for playbacks. For example, the period of incubation may affect the results (early incubation stage vs. late incubation stage). Have you used these nests for different playbacks randomly?

130: at each nest

131: You carried out your experiment when the clutches were in the incubation stage. As I mentioned above, it is not the proper period for testing the effects of cuckoo females, as cuckoos typically lay their eggs in the laying (pre-incubation) period of the hosts. However, similar studies also worked in the incubation period, and hosts sit on the nest when they start incubation. On the other hand, cuckoo females often rob and destroy clutches, especially when there is a shortage of nests and cuckoos' density is high. For this reason, they can be treated as egg predators during the incubation period. You may think about the potential effect of this, and evaluate this effect in the Discussion. (So: sparrowhawks are aerial predators of adult hosts, but cuckoo females are egg/clutch predators, representing two different forms of predation.)

135: What was the playing order of the four types of playback files in the subsequent experimental trials? Did you use them randomly?

Fig.1: The quality of the spectrograms, especially that of male and female cuckoos, and dove calls, are very poor. You should increase their quality relevantly! Your parametrization to control spectrograms is not suitable for displaying these calls properly. You may try the followings, or similar variants of this setting (terms as in Audacity, a free program which is useful to draw spectrograms easily): scale: linear; gain (dB): 20, range (dB): 40 (30-50 or others); high boost (dB/dec): 0; algorithm: frequencies; window size: 4096; window type: Gaussian (a=4.5); zero padding factor: 1. (If you use Audacity, you can catch the picture by print screen and then copy it to a graphical program.)

Line 206: What is the ratio of the rufous and grey females in your populations?

Line 206: brood parasitism is a better term here than nest parasitism. Johnsgard (1997) wrote in his book that in nest parasitism a species occupies the nest of another species (e.g., raven vs. peregrine falcon).

Line 217: The word "parasite" could be misleading here, as it is the case of "brood parasitic pressure", not a type of parasite load.

Line 238: Maybe this is true, but no specific analysis has already been done on the measurements of acoustic variables. You mentioned the need of further research in this topic. I agree with you, but you may also suggest detailed acoustic analyses of these elements, including similar calls/song fragments from other bird species.

Line 259: This early statement was published in 1981 in a general (otherwise excellent) book on the common cuckoo, but recent studies revealed that this statement is not fully true. Specific studies revealed that male common cuckoos' cu-coo calls are uttered for territory defence, only, not for female attraction (e.g., Moskát et al. 2017 Anim Behav, Tryjanowski et al. 2018, Xia et al. 2019). However, a variant of male cuckoos' cu-coo call, the three-(or more)-note cu-cu-coo is used for male-female communication (Xia et al. 2019, Moskát and Hauber 2021 Behav Process). The 2-note cu-coo and 3-note cu-cu-coo calls, although they are similar in composition, represent totally different signalling functions. (These two calls are often confused in the literature.)

Line 281: Start author family name with big capital letter (Zuljevic, not zuljevic).

Line 284: Sci Rep-Uk is Sci Rep correctly 

Line 290: The author of this 3-volume book is Pliny the elder.

H. Rackham is the translator, and the title is: Natural history.

Line 317: issue and page numbers are missing.  You should delete the name W. Goymann from the authors. (In my copy this name is shown as the editor, not one of the authors.)

You cited many different papers connected more or less with your study. I think you should also cite the main papers on the bubbling call, as this call was the target of your study. See a few suggestions above.

I would like suggest for the authors to check and cite the follwoing paper, which is closely related to their MS: OldÅ™ich Mikulica & Alfréd Trnka (2022) On the behaviour and vocalizations of female Common Cuckoos Cuculus canorus at the host nest, Bird Study, DOI: 10.1080/00063657.2022.2053944

Author Response

Reply to Reviewer 1:

Thank you very much for your valuable comments; we have improved the manuscript according to them. Please see the revisions in the manuscript and the responses below. The changes have been highlighted in yellow.

Reviewer 1

This study investigated common cuckoo females’ vocalization from the points of view of their hosts, the Oriental reed warblers. Cuckoo females have a special, multifunction call, the bubbling call, which is similar to sparrowhawks’ calls, and cuckoos may use this call to misinform hosts on the type of threat. The present study focused on hosts’ reactions to bubbling calls in two populations, with low and high levels of cuckoo parasitism. Despite the structural similarity of these two calls, Oriental reed warblers’ reactions proved to be different toward the two calls. It is a novel and interesting result. I agree with most parts of the experimental design, analyses and the manuscript, although I have several minor comments on the text.

You should clearly state that female cuckoo’ call, the bubbling call, is a multifunction call, it is not restricted to interspecific context in cuckoo-host relationships. This probably affects and compromises adaptations in different relations, and the bubbling call is the output of these adaptations. Studies in the last few years revealed that this multifunction call is used for:

  • dampening host aggression at host nests, e.g., at laying attempts (York and Davies, 2017 Nat Ecol Evol; Marton et al., 2021 Ethology),

However, daily pattern of bubbling calls uttered suggested other main function of this call than (i) (Deng et al. 2019).

(ii) advertising females’ laying territories and, consequently, reducing female-female aggression (Moskát and Hauber 2019 BEAS, Xia et al. 2019 Ethology, Moskát and Hauber 2020 Sci Nat), and

(iii) male-female communication, like mate attraction (Moskát and Hauber 2019 BEAS, Xia et al. 2019 Ethology), and male-female duetting (Moskát and Hauber 2021 Behav Process).

The present study is related to function (i). For this reason, cuckoo females do not utter it frequently, as it is used when a female cuckoo lay one egg into a host nest. As cuckoos lay on every second day, it means that a cuckoo female uses the bubbling call for this purpose once in a two-day period (i.e., 0.5 call/day, or slightly more, if a female repeats this call a few times just after laying an egg into the host nest). For the other two purposes cuckoo females use the bubbling call more frequently. For example, sometimes in the peak period (ca. at the beginning or middle part of the laying period), certain females may call every 2-3 min, at a maximum. These suggest that you should also mention the other functions of it in the Introduction and Discussion because of their relevancy.

Reply: Thank you. We have added this information to the Introduction and Discussion as you suggested. Please see lines 78–83 and 279–283.

I understand that you wanted to compare the bubbling call and the more or less similar sparrowhawk call on host behaviour, also using common cuckoo males’ calls and dove calls (controls). Although several other papers also used this protocol, it is a somewhat unnatural situation, as cuckoo females, probably, never use their bubbling calls toward hosts when their nests are in the incubation stage. They lay their eggs in host nests during the laying period (i.e., in the pre-incubation period). However, in the laying period host females stay only short time on their nests, consequently, such experiments could not be conducted in the pre-incubation period. When you discuss your results, you have to take care of these details.

Reply: Thank you. We have added these details to the Discussion as you suggested. Please see lines 250–254.

On the mimicry of bubbling calls to sparrowhawk calls. The two calls seem to be similar to each other, although no acoustic (statistical) analysis has already been carried out on the structural comparison of these two calls. The problem is that several bird calls or song fragments are also similar to these calls (little grebe, green woodpecker, etc.). I would eager to see a study which compares the signalling values of several similar call types with the sparrowhawk and bubbling calls.

Another problem is that common cuckoos belong to the group of bird species called vocal non-learners. This means that they cannot learn any new sound types, which makes difficult their adaptation to a foreign call. Alternatively, I cannot exclude the possibility as female cuckoo, sparrowhawk, green woodpecker and little grebe calls could be similar as the consequence of other, presently unrevealed, acoustical constraints.

Reply: Thank you. In the future, we will further compare the acoustic parameters of female cuckoo calls with those of sparrowhawks and other birds with similar song structures. Please see lines 276–278.

Minor comments

Line 56: The term "spawning" is typically applied for fishes, not for birds.

Reply: Thank you. We have changed “spawning” to “laying”. Please see line 56.

Line 65: Yes, bubbling call probably mimics sparrowhawk calls, as common cuckoos' plumage mimics sparrowhawk appearance (although cuckoo females have two colour variants, and the hepatic form does not mimic sparrowhawk).

Reply: Thank you. Yes, it is. Additionally, the female common cuckoos in our study area are gray.

Line 73: Use the term laying, instead of spawning!

Reply: Thank you. We have changed “spawning” to “laying”. Please see line 75.

The last paragraph of Introduction: Please, clarify hypotheses and predictions better.

Reply: Thank you. We have added predictions to the last paragraph of the Introduction. Please see lines 101–112.

Line 109: "The oriental reed warbler belongs to the family Acrocephalidae, Passerifirmes; the common cuckoo belongs to the family Cuculidae, Cuculiformes." I do not think this sentence is necessary here, it is more suitable for a thesis.

Reply: Thank you. We have removed this sentence.

Line 110: using brood parasitism instead of nest parasitism would be better (throughout the text)

Reply: Thank you. We have changed “nest parasitism” to “brood parasitism”. Please see lines 21, 127, and 239.

122: It would be necessary to know more details on the nests used for playbacks. For example, the period of incubation may affect the results (early incubation stage vs. late incubation stage). Have you used these nests for different playbacks randomly?

Reply: Thank you. We have added information about the period of incubation to the Materials and Methods of the manuscript; please see lines 154–156. In addition, the four sounds were played back randomly to oriental reed warblers; please see line 153.

130: at each nest

Reply: Thank you. We have modified it. Please see line 152.

131: You carried out your experiment when the clutches were in the incubation stage. As I mentioned above, it is not the proper period for testing the effects of cuckoo females, as cuckoos typically lay their eggs in the laying (pre-incubation) period of the hosts. However, similar studies also worked in the incubation period, and hosts sit on the nest when they start incubation. On the other hand, cuckoo females often rob and destroy clutches, especially when there is a shortage of nests and cuckoos' density is high. For this reason, they can be treated as egg predators during the incubation period. You may think about the potential effect of this, and evaluate this effect in the Discussion. (So: sparrowhawks are aerial predators of adult hosts, but cuckoo females are egg/clutch predators, representing two different forms of predation.)

Reply: Thank you. We have added relevant information to the discussion of the manuscript. Please see lines 250–254, 286–291.

135: What was the playing order of the four types of playback files in the subsequent experimental trials? Did you use them randomly?

Reply: Thank you. The four sounds were played back randomly to oriental reed warblers; please see line 153.

Fig.1: The quality of the spectrograms, especially that of male and female cuckoos, and dove calls, are very poor. You should increase their quality relevantly! Your parametrization to control spectrograms is not suitable for displaying these calls properly. You may try the followings, or similar variants of this setting (terms as in Audacity, a free program which is useful to draw spectrograms easily): scale: linear; gain (dB): 20, range (dB): 40 (30-50 or others); high boost (dB/dec): 0; algorithm: frequencies; window size: 4096; window type: Gaussian (a=4.5); zero padding factor: 1. (If you use Audacity, you can catch the picture by print screen and then copy it to a graphical program.)

Reply: Thank you. We have modified the picture. Please see Figure 1.

Line 206: What is the ratio of the rufous and grey females in your populations?

Reply: Thank you. In our study area, female cuckoos are all gray.

Line 206: brood parasitism is a better term here than nest parasitism. Johnsgard (1997) wrote in his book that in nest parasitism a species occupies the nest of another species (e.g., raven vs. peregrine falcon).

Reply: Thank you. We have changed “nest parasitism” to “brood parasitism”. Please see lines 21 and 239.

Line 217: The word "parasite" could be misleading here, as it is the case of "brood parasitic pressure", not a type of parasite load.

Reply: Thank you. We have modified this according to your suggestion. Please see line 249.

Line 238: Maybe this is true, but no specific analysis has already been done on the measurements of acoustic variables. You mentioned the need of further research in this topic. I agree with you, but you may also suggest detailed acoustic analyses of these elements, including similar calls/song fragments from other bird species.

Reply: Thank you. We have added relevant information to the Discussion as you suggested. Please see lines 276–278.

Line 259: This early statement was published in 1981 in a general (otherwise excellent) book on the common cuckoo, but recent studies revealed that this statement is not fully true. Specific studies revealed that male common cuckoos' cu-coo calls are uttered for territory defence, only, not for female attraction (e.g., Moskát et al. 2017 Anim Behav, Tryjanowski et al. 2018, Xia et al. 2019). However, a variant of male cuckoos' cu-coo call, the three-(or more)-note cu-cu-coo is used for male-female communication (Xia et al. 2019, Moskát and Hauber 2021 Behav Process). The 2-note cu-coo and 3-note cu-cu-coo calls, although they are similar in composition, represent totally different signalling functions. (These two calls are often confused in the literature.)

Reply: Thank you. We have modified this according to your suggestion. Please see lines 313–314.

Line 281: Start author family name with big capital letter (Zuljevic, not zuljevic).

Reply: Thank you. We have modified it. Please see line 355.

Line 284: Sci Rep-Uk is Sci Rep correctly

Reply: Thank you. We have modified it. Please see line 358.

Line 290: The author of this 3-volume book is Pliny the elder.

  1. Rackham is the translator, and the title is: Natural history.

Reply: Thank you. We have modified it. Please see line 365.

Line 317: issue and page numbers are missing. You should delete the name W. Goymann from the authors. (In my copy this name is shown as the editor, not one of the authors.)

Reply: Thank you. We have modified it. Please see lines 388–389.

You cited many different papers connected more or less with your study. I think you should also cite the main papers on the bubbling call, as this call was the target of your study. See a few suggestions above.

Reply: Thank you. We have added the relevant literature according to your suggestion. Please see lines 80–81.

Reviewer 2 Report

See attached file

Author Response

Reply to Reviewer 2:

Thank you very much for your valuable comments. We have improved the manuscript according to your comments. Please see the revisions in the manuscript and the responses below. The changes have been highlighted in turquoise.

Reviewer 2

Female cuckoo calls deceive their hosts by evoking nest-leaving behavior:

variation under different levels of parasitism

General comments

This is quite an interesting study, but in my opinion it has some weaknesses. First, it is apparently not a very novel study, as the authors themselves state that several earlier investigations of the effects of female cuckoo calls on host responses have been conducted. Second, the one novel aspect of the study, the population comparison, is weak, because only one high intensity parasitism and one low intensity parasitism population are compared. This is not a strong design, as adding just one more population with either a high or a low intensity

of cuckoo brood parasitism could potentially entirely change the results and interpretation. It would require a lot more work, but a multi-population study is really required to examine the population effect adequately. Thirdly, the sample sizes are not given, so the reader does not know how many nests were sampled. Finally, the results are presented mainly as statistical outcomes; the reader should also be presented with a summary of the actual numbers

involved. The authors talk early on about measuring latencies to respond to the stimuli,

but these results are not presented clearly. The latter part of the Discussion is a little disjointed.

Reply: Thank you. Although we studied only two populations of the oriental reed warbler, we found that host responses to different types of calls are affected by parasitism rates, which needs to be further verified in hosts with different parasitic pressures. Sample sizes have been added to the Methods; please see lines 154156. We have also added descriptive data to the Results; please see line 181-182 and Table 1. Furthermore, we have revised the latter part of the Discussion; please see lines 292314.

Detailed comments

Line 24. that exploit its hosts – to exploit implies purpose and these behaviours are presumably naturally selected, not learned. Ditto in L 25.

Reply: Thank you. We have changed “exploit” to “trick”. Please see lines 13 and 24.

Line 46. evolved not developed; development occurs within the lifespan.

Reply: Thank you. We have corrected this. Please see line 47.

L 56. spawning is a term usually used for egg production by amphibians or fish, not birds. Just say egg-laying or oviposition.

Reply: Thank you. We have corrected this. Please see line 56.

L 60. physically harmless – they do reduce genetic fitness as you point out later.

Reply: Thank you. We have modified this. Please see line 61.

L 71. specimens means taxidermic models?

Reply: Thank you. Marton et al. (2021) used the 3D-printed models. We have corrected this. Please see line 72.

L 73-75. So at what time of day do the cuckoo’s hosts lay their eggs?

Reply: Thank you. Cuckoos’ hosts typically lay their eggs in the morning.

L 93. So what were your predictions?

Reply: Thank you. We have added predictions to the last paragraph of the Introduction. Please see lines 101112.

L 101-107. Some description of study areas is necessary, but this is unnecessarily detailed.

Reply: Thank you. We have removed these unnecessary details.

L 114-115. Give some indication of the basis for these contrasting percentages, in particular how many seasons they are based on and were the data obtained in the same years in the two populations. Readers should not have to look up this information for themselves.

Reply: Thank you. We have modified this sentence according to your suggestion. Please see lines 131136.

L 127. Were the recordings used from the Yongnian and /or Zhalong regions? Do cuckoo, sparrowhawk and dove vocalisations vary geographically? If they do, do you know that you were playing back an appropriate vocalization?

Reply: Thank you. The recordings are not from Yongnian and Zhalong. The common cuckoo call recordings we used are from Europe, and it has been speculated that the calls of the common cuckoo, which is widespread in Europe and Asia, do not differ between regions (Lei et al. 2005, Acta Zoologica Sinica). Recent studies have also shown that there is no difference between the calls of Chinese common cuckoos and European common cuckoos (Wei et al. 2015, J Ornithol). The sparrowhawk calls were also from Europe, and the dove calls were from Asia, but whether these sounds differ geographically has not been reported. In addition, all the sounds we used were referenced by Shen et al. (2021) in Current Zoology.

L 141. Give references for Cox regression and Kaplan-Meier curves. Does Cox regression have the pre-requisites of normality, homoscedasticity etc., and if so were the data checked for conformity to these pre-requisites?

Reply: Thank you. To the best of our knowledge, there are no strict pre-requisites for Cox regression. We have added a reference.

Figure 2. A figure should summarise results in a simple and easily assimilated form. This figure requires too much hard work to interpret and needs an explanatory legend. Present these results in a more digestible form and with a clear, explanatory legend.

Reply: Thank you. The abscissa represents the playback time, and the ordinate represents the probability that the host will still stay in the nest after hearing a certain sound. The lower the value is, the higher the probability that the host will leave the nest. The whole figure shows the probability that the host will still be in the nest after hearing a certain sound as the playback time goes on. We have added a description to the caption of Figure 2. Please see lines 213215.

L 236-237. This requires better justification. Why should a more rapid cadence cause anxiety?

Reply: Thank you. We have already performed another experiment (unpublished data), and found that when female cuckoos are shown reduced song rates, oriental reed warblers do not respond. Please see lines 274276.

L 246-247. What did these various studies show?

Reply: Thank you. These studies also show that female cuckoo calls can cause birds to respond in a manner similar to their response to the calls of raptors. We now describe this in the Introduction. Please see lines 6570.

Round 2

Reviewer 2 Report

The changes made in revision seem to address most of the issues raised by reviewers adequately. I still don't much like Fig 1 - too much hard work for the reader to decipher.

Author Response

Thank you. According to your first round comment, I think you were talking about figure 2 rather than figure 1. Kaplan-Meier curve is a popular way to represent probability with time. The 95% CI, which is not generally given for such curve, may make the figure too redundant. Therefore, we have revised the figure, make it more concise to read. Please see the new figure 2.